# GRAM: Graph Regularizable Assessment Metric

*******1[0000−1111−2222−3333] and *******2[1111−2222−3333−4444]

1 **********, ******, ***
2 *******, ********. **, ***** *****, ****** ******@*****.com

**Abstract.** Here, we propose the Graph Regularizable Assessment Metric ($GRAM$), a customizable tool for evaluating the quality of generated brain graphs. Current geometric deep learning methods often lack robust quantification techniques for assessing the synthetic brain graphs integrity. $GRAM$ addresses this gap by proportionally combining a set of existing graph metrics to establish a linear correlation between applied distortions and ground-truth graphs. To evaluate the performance of our model, we generated a synthetic dataset of structural brain connectomes which was derived from an existing dataset and used to simulate a set of predicted connectomes from a generative model with controlled levels of distortions. Our results show that $GRAM$ outperforms single metrics in quantifying the distortion between generated and original graphs. This approach is a significant step towards establishing a universal graph quality index for graph-based predictive studies.

**Keywords:** Predicted brain graphs · Quality metrics · Customized metrics.

## 1  Introduction

Brain connectomes are crucial for exploring the connectivity patterns underlying cognitive processes [21]. These connectomes provide a framework for predicting the progression of neurodegenerative diseases by integrating connectomic analyses with established neuroscientific knowledge [4]. For instance, [10,13] introduced geometric deep learning approaches to forecast Alzheimer's disease progression using brain connectome data. Despite their potential, obtaining connectomic data poses significant challenges. One major impediment is the extensive processing required for neuroimages acquired through modalities such as Magnetic Resonance Imaging (MRI). This process is both time-consuming and computationally intensive. Another challenge is the limited availability of sufficient MRI data, which can hinder comprehensive analyses.

One approach to address the challenge of limited MRI data involves the use of generative models to produce synthetic neuroimages. Generative Adversarial Networks (GANs) [7] have shown significant capability in generating realistic brain scans. For instance, [14] proposed a GAN model based on a fully convolutional network and an Auto-Context Model to enhance the realism and accuracy of synthetic images. Similarly, [19] developed a GAN model that produces

high-quality, realistic images that simulate the ground-truth brain images- to improve the performance of diagnostic models in medical diagnostics. Despite the potential benefits, using GAN-generated MRI data to study brain connectomes introduces two key challenges. First, the generated MRI data must be indistinguishable from real data both quantitatively and qualitatively. Second, the synthetic data requires additional processing to extract connectivity matrices.

To address the above-mentioned issues, [25] proposed predicting brain connectivity matrices using a graph GAN-based approach. The authors created representative templates from clustered brain graphs to train models that predict the evolution of connectivities for a given brain disease over time. Their novel few-shot learning framework uses minimal training data and employs clustering and Connectional Brain Templates (CBTs) to handle the diversity within brain connectomic data. This ensures robust model training despite limited data. However, unlike images, brain connectomes are virtually impossible to evaluate qualitatively. Instead, quantitative metrics (e.g., centrality measures) are, *thus far*, a single way to evaluate the quality of the generated graphs.

In this paper, we highlight the limitations of existing metrics for graph quality assessment and propose a novel universal customizable metric to quantify the quality of generated graphs with an application to a simulated prediction of brain connectivities based on an existing dataset. In particular, we propose Graph Regularizable Assessment Metric ($GRAM$), a customizable framework designed to learn to proportionally combine a set of existing graph metrics in order to evaluate the generated graph's quality. Drawing inspiration from the universal image quality index by [23], $GRAM$ could be considered a first step towards a more universal graph quality index. Our contributions are listed as follows:

1. We propose a new general assumption for quantitatively interpreting the quality of a generated graph based on the linearity between the amount of distortion and the value of the reported metric.
2. We propose a novel general metric based on the weighted combination of existing metrics.
3. Our proposed metric is adjustable depending on the type of graph as well as the chosen metrics to report.

## 2   Methods

In this section, we present in detail the proposed metric $GRAM$ for quantifying the quality of generated graphs.

### 2.1   Simulation of Generated Brain Graphs

Let $G = (V, E)$ be a directed weighted graph, $V$ denotes the vertices, and $E$ denotes the weighted edges given by $w : e \to \mathbb{R}$. Let $\hat{G}$ be the simulation of the output of a given generative model $F$ aiming to predict a target brain connectome $G$ such that $\hat{G} \approx G$. The goal of the simulation is to bypass the problem of finding the optimal $F$ to train as well as to control the amount of distortion $d$ between $\hat{G}$ and $G$, where $d \in ]0, 1]$ with $s$ defined as distortion step.

The distortion level between $\hat{G}$ and $G$ is measured by the number of edges $|E|$ with differing weights. Specifically, for an edge $e \in E$ with weight $w$ in $G$

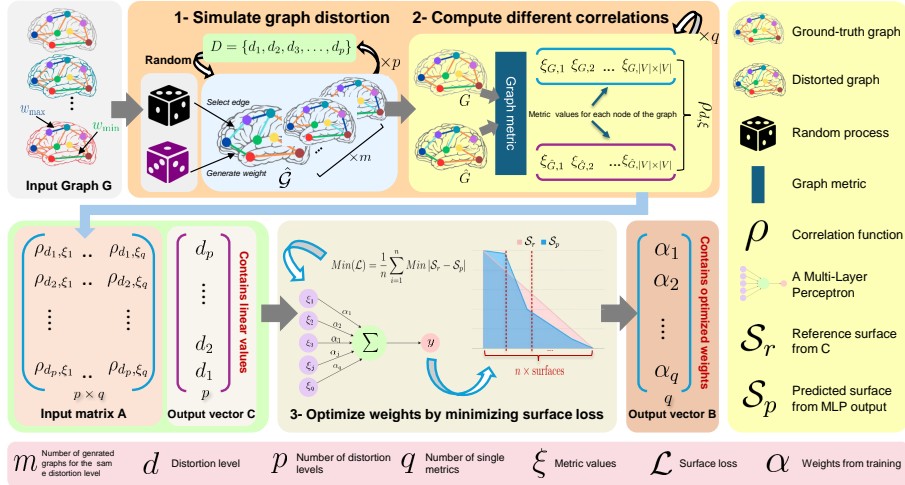

**Fig. 1.** *Pipeline of the proposed GRAM metric for assessing the quality of directed weighted graphs.* **(1) Simulate graph distortion.** For an input graph $G$ and for a distortion level $d$ we alter weights of randomly selected edges by random generated values in $[w_{\min}, w_{\max}]$ producing $m$ distorted graphs. **(2) Compute different correlations.** First we apply the single metrics to the ground truth graph and the distorted one then calculate the Pearson, Spearman and Kendall's Tau correlations between them. Second for each graph and each correlation coefficient, we generate a matrix $A$ of size $(p \times q)$ containing the correlation values organized by distortion levels vertically and single metrics horizontally. **(3) Optimize weights by minimizing surface loss.** We train $GRAM$ using an MLP to optimize the metrics' weights $\alpha_j$ forming the vector $B(q)$ by minimizing the loss between the predicted surface created by the MLP output $A \times B$ and the reference surface created by the vector $C(p)$ across $n$ surfaces.

and $\hat{w}$ in $\hat{G}$, the distortion is defined as the proportion of edges for which $w \neq \hat{w}$, regardless of the magnitude of the difference $|w - \hat{w}|$.

As shown in Fig. 1, we define a set of $m$ distorted graphs, each characterized by a distortion level $d$ denoted as $\hat{\mathcal{G}}_d = \{\hat{G}_1, \ldots, \hat{G}_m\}$. The process of generating a suite of distorted graphs across all distortion levels is detailed in Algorithm 1, resulting in the set $\hat{\mathcal{G}} = \{\hat{\mathcal{G}}_s, \ldots, \hat{\mathcal{G}}_1\}$. Initially, we set a predefined distortion step (e,g., 0.1). For each increment of the distortion level $d$, we randomly select $|\hat{E}|$ edges, where $|\hat{E}| = d \times (|E| - |V|)$. Here, $|E|$ is the total number of edges in the graph, and $|V|$ is the total number of vertices. The term $|E| - |V|$ represents the total number of non-diagonal edges, as diagonal edges (self-loops) are excluded. Therefore, by subtracting $|V|$ from $|E|$, we ensure that we only consider non-diagonal edges. At each selected edge $e$, we replace its weight with a randomly generated value within the range $[w_{\min}, w_{\max}]$, where $w_{\min}$ and $w_{\max}$ are the minimum and maximum weight values in the graph $G$, respectively. We repeat this process $m$ times to ensure that all the edges are distorted at least once.

---

**Algorithm 1** Generate Distorted Graphs

---

**Require:** Directed weighted graph $G = (V, E)$, distortion step $s$, number of iterations $m$

**Ensure:** Set of distorted graphs $\hat{\mathcal{G}}$

1: Initialize $\hat{\mathcal{G}} \leftarrow \emptyset$
2: $w_{\min} \leftarrow \min\{w(e) \mid e \in E\}$
3: $w_{\max} \leftarrow \max\{w(e) \mid e \in E\}$
4: **for** $d$ in $D$ with step $s$ **do**
5:      Initialize $\hat{\mathcal{G}}_d \leftarrow \emptyset$
6:      $|\hat{E}| \leftarrow d \times (|E| - |V|)$
7:      **for** $i$ from 1 to $m$ **do**
8:          $\hat{G} \leftarrow G$
9:          Select $|\hat{E}|$ random edges from $E$
10:         **for** each selected edge $e$ **do**
11:             $\hat{w}(e) \leftarrow \mathrm{random}(w_{\min}, w_{\max})$
12:             Update edge weight in $\hat{G}$ to $\hat{w}(e)$
13:         **end for**
14:         Add $\hat{G}$ to $\hat{\mathcal{G}}_d$
15:     **end for**
16:     Add $\hat{\mathcal{G}}_d$ to $\hat{\mathcal{G}}$
17: **end for**
        return $\hat{\mathcal{G}}$

---

## 2.2   Graph Reliability Assessment Metric (GRAM)

**Assumption 1:** Let $G = (V, E)$ represent a brain graph, where $V$ denotes vertices and $E$ denotes edges, with $w : e \to \mathbb{R}$ representing the weights of the edges in $E$. For a metric $\mathcal{M}$ that assesses graph quality, we postulate that the variation in $y$ such that:

$$y = \rho(\mathcal{M}(G), \mathcal{M}(\hat{G})) \tag{1}$$

is **linearly correlated** with the distortion $d$ applied to $G$. Where $\hat{G}$ is the distorted graph, and $\rho$ is the correlation function. We express $\mathcal{M}$ as follows:

$$\mathcal{M}(\hat{G}) = 1 - k \times d \tag{2}$$

Where $d$ is the distortion level expressed as a ratio (e.g., $d = 0.1$ for 10% distortion), and $k$ is a constant scaling factor.

We introduce the $GRAM$: an adjustable and learnable measure for evaluating the quality of generated graphs. Unlike existing metrics, such as centrality measurements that separately assess different graph aspects, $GRAM$ provides a linear approximation of the relationship between distortion evolution and its output value taking into consideration multiple aspects of the graph. For instance, a $GRAM$ value of 0.8 indicates that the graph is 80% similar to the original data. The result of our metric is represented by the $y$ value, which indicates the degree of similarity between the generated and original graphs.

To do so, as seen in Fig. 1 for each graph, we define a matrix $A \in \mathbb{R}^{p \times q}$, such that $p$ is the number of distortion levels, $q$ is the number of existing metrics.

Within $A$, each element $A_{i,j}$ represents the correlation between a given metric's output $\xi(G)$ and $\xi(\hat{G})$ applied to $G$ and $\hat{G}$, respectively, at a particular distortion level $d$. Here $i$ indexes a distinct distortion level, and $j$ refers to a particular metric correlation (e.g., At a distortion increment $s = 0.1$, $A_{1,3}$ corresponds to the third metric correlation at a 10% distortion level). We define $C$ as the reference output of the metric ensuring adherence to Assumption 1. $GRAM$ aims to find the values $\alpha_j$ forming a vector $B$ such that: $A \times B = C$.

We define $GRAM(G)$ as a weighted sum of the metrics' correlations between the ground truth and distorted graphs. Specifically, let $\xi_j(G)$ and $\xi_j(\hat{G})$ denote the $j$-th metric evaluated on $G$ and the distorted graph $\hat{G}$, respectively. Then:

$$GRAM(G) = \sum_{j=1}^{q} \alpha_j \times \rho(\xi_j(G), \xi_j(\hat{G})) \tag{3}$$

Where $\rho$ is the correlation function and $\alpha_j$ are the learnable weights for each metric's correlation.

To solve the vector $B$, we leverage the universal approximation theorem demonstrated by [8], which establishes that a feedforward neural network featuring a single hidden layer can approximate any continuous function with sufficient neurons and appropriate parameters (weights and biases). Consequently, our approach involves training a Multi-Layer Perceptron (MLP) to determine the parameters within $B$, taking $A$ as input and $C$ as output.

To train the MLP, we minimize the loss between two surfaces: $S_r$, formed by the intersection of vector $C$ with the $X$ and $Y$ axes, and $S_p$, defined by the curve of predicted weights in $B$, where the vector $A \times B$ intersects the $X$ and $Y$ axes, Fig. 1 (3). The goal is to optimize the predicted weights in $B$ so that the vector $A \times B$ closely approximates vector $C$, aligning surfaces $S_r$ and $S_p$.

To do so, we use least squares regression [11] to fit a polynomial function

$$P(x) = a_n x^n + a_{n-1} x^{n-1} + \cdots + a_1 x + a_0$$

to the $A \times B$ output data. This involves finding a polynomial that minimizes the sum of the squared differences between the MLP output data points and the polynomial's predicted values. This method creates a continuous curve that closely follows the data pattern formed by $A \times B$, which we then use to approximate the integral within a specified range.

In our study, the loss is minimized by the following process: first, the total surface created by the MLP values is divided into $n$ distinct parts. Each part is optimized independently to simplify parameter convergence. Finally, we average all the results from the optimizations.

Our proposed surface loss can be defined as follows:

$$\text{Surface Loss} = \frac{1}{n} \sum_{i=1}^{n} \int_{x_{\min}}^{x_{\max}} |f_C(x) - f_{\text{MLP}}(x)| \, dx \tag{4}$$

where $n$ represents the number of distinct surface parts, $x_{\min}$ and $x_{\max}$ denote the minimum and maximum values of the input range, respectively. The function

$f_C(x)$ denotes the line defined by $C$ values, while $f_{\mathrm{MLP}}(x)$ corresponds to the polynomial approximation function of the MLP output.

**Training details.** We train our model for 250 epochs, using Google Colab. For optimization, we use Adam optimize [9], with learning rate of 0.01.

## 3    Results and discussion

In this section, we evaluate 10 selected individual graph metrics as well as our proposed $GRAM$. Additionally, we discuss each of the findings.

### 3.1    Dataset

We used a dataset from [20] that contains 88 subjects (48 females, 40 males aged between 18 and 48 years). All subjects are right-handed and healthy. The dataset contains the structural connectomes where each connectome contains 90 brain regions of intrest from the Automated Anatomical Labeling Atlas (AALA) [22].

### 3.2    Single Metric Evaluation

For this study, we select a step of $s = 0.1$ and a set of ten widely utilized graph metrics in the literature [16]. These metrics are: Betweenness Centrality [6], Closeness Centrality [17], Weighted Degree Centrality [1], Eigen Centrality [2], Pagerank Centrality [15], Katz Centrality [3], Hub-Authority [5], Harmony [12], Average Neighbor Degree [24], Diversity Index [18]. As a baseline for evaluating our proposed $GRAM$, Fig. 2, displays the correlation between the ground truth graphs and the generated ones, for each individual metric across various levels of distortion.

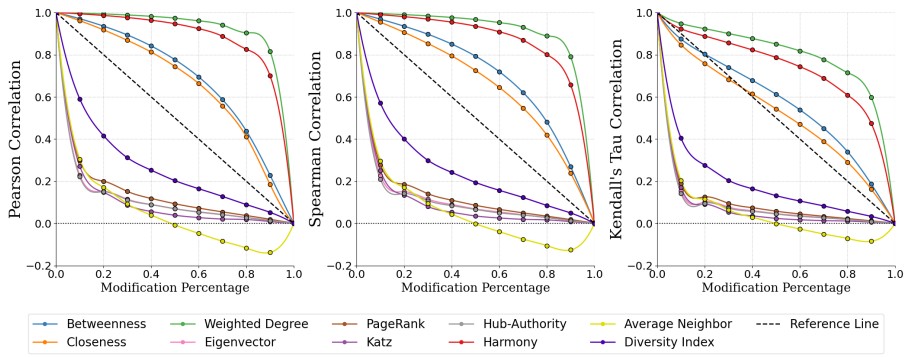

**Fig. 2.** *Correlations for single metrics.* We plot different correlations (Pearson, Spearman and Kendall's Tau correlations) between the ground truth graphs and the distorted ones, for each individual metric across various levels of distortion.

Fig. 2 shows the evolution of the independent metrics differs across the studied correlation coefficients. These evolutions are non-linear and could be visually

categorized into two distinct patterns. The first pattern includes metrics such as Betweenness, Closeness, Harmony, and Weighted Degree Centralities which exhibit a moderate progression for values of $d < 0.8$ followed by a rapid decline towards a correlation values of 0. Contrarily, the second pattern shows metrics such as Eigenvector, PageRank, Katz, and Diversity Index. These metrics show a non-linear evolution, characterized by a rapid correlation decline for distortion levels lower than $d = 0.3$. This decline is then followed by a gradual stabilization of the correlations for levels where $d > 0.3$. This observation highlights the insufficiency of relying on a single set of metrics to comprehensively assess generated graph quality. All metrics exhibit non-linear correlations compared to the reference line, thus rendering them unreliable due to their *under-estimation* or *over-estimation* of distortion levels.

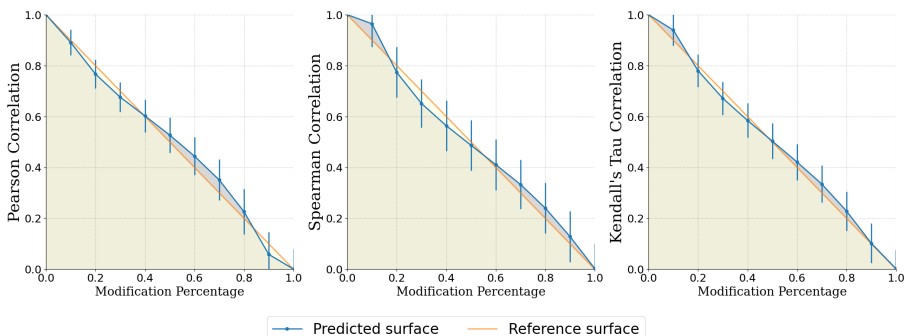

**Fig. 3.** *GRAM testing results.* The figure illustrates the intersection of $A \times B$ (blue) and refrence vector $C$ (orange) as surfaces intersecting the $x$ and $y$ axes, shown for Pearson, Spearman, and Kendall's Tau correlation coefficients.

### 3.3 GRAM evaluation

We generated distorted graphs using a step value of $s = 0.1$ and trained $GRAM$ using the ten previously listed metrics. The training process optimizes two separate surfaces. The loss for the first surface is calculated over the range $[0.1, 0.5]$, while the loss for the second surface is calculated over the range $[0.5, 1]$. Fig. 3 shows $GRAM$ testing results based on the previously mentioned correlations (i.e., Pearson correlation coefficient, etc ...). Visually, the correlational outputs of $GRAM$ seem to closely approximate the target triangular shape created by the reference line and its intersection with $x$ and $y$ axis.

Table 1 shows the weight of each metric as produced by $GRAM$ across various correlation coefficients. The average neighbor degree consistently exhibits high weight values across Pearson, Spearman, and Kendall's Tau correlation coefficients. Yet, almost all the other metrics' weights are close to $(0 \pm 0.1)$. This disparity in the correlation's values may be due to the significant overlap in the information captured by the average neighbor degree and closeness centrality

with other metrics like betweenness or diversity index. The computation redundancy in some metrics could lead to one of these metrics to be over-represented compared to similar metrics.

| $\alpha_j$ | $\rho_p$ | $\rho_s$ | $\rho_k$ | Single metrics |
|:---:|:---:|:---:|:---:|:---|
| $\alpha_1$ | 0.230 | 0.250 | 0.643 | Betweenness |
| $\alpha_2$ | 0.473 | 0.159 | 0.166 | Closeness |
| $\alpha_3$ | -0.087 | 0.223 | 0.106 | Weighted Degree |
| $\alpha_4$ | -0.067 | 0.056 | 0.033 | Eigenvector |
| $\alpha_5$ | 0.133 | 0.024 | 0.129 | PageRank |
| $\alpha_6$ | 0.022 | 0.069 | 0.006 | Katz |
| $\alpha_7$ | 0.001 | 0.006 | -0.059 | Hub-Authority |
| $\alpha_8$ | 0.022 | -0.159 | -0.156 | Harmony |
| $\alpha_9$ | 0.309 | 0.614 | 0.578 | Average Neighbor Degree |
| $\alpha_{10}$ | 0.260 | 0.501 | 0.349 | Diversity Index |

**Table 1.** Optimised $\alpha_j$ values for Pearson $\rho_p$, Spearman $\rho_s$, and Kendall's Tau $\rho_k$ correlations

**Limitations and Future Directions.** This study marks an initial effort to establish a universal metric for assessing the quality of generated brain graphs. One notable limitation is the selection of ten metrics that share similar calculation methods. Another limitation is the evaluation based on a single dataset. Future research should explore a broader range of metrics and evaluate the model across various graph datasets for different applications.

**Code Availability.** All codes used for this study are available in: `https://shorturl.at/xjSPW`

## 4   Conclusion

This paper introduced the Graph Regularizable Assessment Metric ($GRAM$) to evaluate the quality of generated brain graphs that could be used as a universal method in reporting the quality of generated graphs in future predictive studies. It combines multiple metrics in a weighted framework, addressing the limitations of existing graph quality metrics. Our proposed method establishes a general assumption for graph quality based on the linearity between distortion and metric values where we used a multi-layer perceptron to optimize metric weights. We test $GRAM$ using a set of simulated structural connectome data on which it demonstrated reasonable reliability in quantifying graph quality. In future work, we aim to extend $GRAM$ to diverse graph types and datasets.

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
