# OpenReview forum: "GRAM: Graph Regularizable Assessment Metric"
_MICCAI.org/2024/Workshop/MSB — MICCAI Student Board EMERGE Workshop 2024 Oral_

### Official Review · Reviewer_fUna · 2024-07-05

**Recommendation:** 5
**Confidence:** 5

**Clarity:**

The paper is clear and well-written, with minor areas for improvement in clarity

**Feedback:**

1. Larger dataset and more benchmark evaluation:
   - Evaluate bigger and more diverse datasets (age ranges, health conditions). What happens if datasets are imbalanced for a certain condition (which is often the case for neuroimaging datasets)?
   - It would also be interesting to compare GRAM's performance on functional connectomes.

2. Broaden the range of graph metrics (as acknowledged by the authors)
   - Include global network properties and neuroscience-specific metrics.
   - Explore different combinations of metrics => Could certain combinations be best suited for specific downstream tasks/connectome types/pathological conditions?

3. It would be great to see how GRAM compares to existing graph similarity measures.

4. It would also be interesting to test GRAM on graphs from actual predictive models (e.g., graph GANs, VAEs).

5. Due to the simulation method, there seems to be little/no discussion demonstrative of potential impact on clinical applications.

6. Regarding GRAM properties, the authors should conduct sensitivity analyses and investigate performance on different graph types (e.g., parcellations) and sizes (varying m).

7. For the results shown, I would suggest the authors perform formal statistical tests comparing GRAM to other methods.

8. Another unaddressed limitation is GRAM's computational complexity compared to existing metrics, which are often quick to run within existing libraries and pipelines. As such, it would be useful for the paper to report on the computational cost and, if not high, what was done to ensure that.

9. I also note concern about GRAM potentially overfitting. The authors don't disclose the method used for the train/validation/test splits. I would suggest implementing cross-validation and testing on a held-out dataset.

10. Lastly, future work addressing the ethical implications and recommendations of using AI-generated brain graphs in clinical settings would also be interesting.

**Justification:**

Overall, this was a really good paper with clear novel contributions and strong performance. There are some limitations of concern regarding the constraints and potential scalability/generalizability based on the training dataset used & the single graph metrics.

**Reproducibility:**

Sufficient amount of details available for reproducing the main results, but open access is not provided to source code and/or data

**Strengths:**

1. GRAM combines multiple graph metrics into a single, customizable framework and establishes a linear, interpretable relationship between distortion levels and metric values.
2. The simulation method for distorting brain graphs is implemented and assessed systematically across different distortion levels without needing a real generative model.
3. To optimize performance, The paper thoroughly assesses single metrics and GRAM across different distortion levels and correlation types using a comprehensive surface loss.
4. While primarily focused on brain graphs, GRAM can also be useful in related domains where graph generation or prediction is important.
5. The paper provides a much-needed tool for quantitatively assessing the quality of generated brain graphs.

**Summary:**

The paper proposes GRAM, a Graph Regularizable Assessment Metric for evaluating the quality of synthetically generated brain graphs. The authors test GRAM's performance over 10 commonly used graph metrics and noted improved qualitative and quantitative meaningfulness.

**Weaknesses:**

1. The study uses a small dataset of 88 subjects from a specific age range (18-48 years) and health status (all healthy). This limits the generalizability of the results. It's unclear how GRAM could scale to larger populations or how it would perform on datasets with different characteristics or pathological conditions.

2 The authors acknowledge that the ten chosen graph metrics share similar calculation methods. It would be interesting to evaluate GRAM on metrics that capture global network properties or are more specific to neuroscience (e.g., rich club coefficient). For example, see "Rich-club in the brain’s macrostructure: Insights from graph theoretical analysis" by Kim and Min (2020) and "Trajectory of rich club properties in structural brain networks" by Riedel et al. (2022).

3. While the paper compares GRAM to individual metrics, it doesn't compare it to other existing methods for assessing graph quality or similarity, such as graph edit distance or graph kernel methods. This makes it difficult to assess GRAM's advantages over current approaches fully.

4. The evaluation relies entirely on simulated distortions of real brain graphs. While this provides controlled conditions for testing, it may not accurately represent the types of differences that would occur in graphs generated by actual predictive models. Testing on graphs produced by real generative models would strengthen the validation.

5. The paper doesn't demonstrate how improvements in graph quality assessment translate to clinical applications. Discussing or demonstrating how GRAM could impact diagnosis or prognosis in neurological disorders would strengthen the paper's relevance.

6. The paper doesn't discuss/explore how GRAM's performance changes with different combinations of metrics or how it performs on different types of graphs (e.g., functional vs. structural connectomes).

7. The paper doesn't discuss GRAM's computational cost compared to individual metrics. If GRAM is significantly more computationally expensive, its practical applicability could be limited.

9. Using an MLP to optimize metric weights introduces the possibility of overfitting the specific dataset used (especially since it's so small and constrained). The paper doesn't address how this could be mitigated or how well GRAM would generalize to unseen data.

10. There were a few typos in the paper, e.g. "genrated" in Fig 1 and "refrence" in Fig. 3 caption. The bottom pink banner for Fig 1 also has some distortion where text runs over weirdly. Would adjust the font size/words used.

---

> ### Author Response · Authors · 2024-07-14
> **Rebuttal by Authors**
>
> We thank the reviewer for the significant effort they made to provide feedback for this manuscript. We addressed each feedback as follows:
>
> Feedback 1: Small dataset limitation.
> Response 1: We appreciate the reviewer's observation. In our study, we generated 100 variations for each graph across all levels, resulting in a total of 8,800 graphs. We apologize for any previous lack of clarity on this point and have updated the manuscript to explicitly state: "We emphasize that our simulations resulted in a total of 8,800 graphs."
>
> Feedback 2: Evaluate GRAM on metrics that capture global network properties or are more specific to neuroscience (e.g., rich club coefficient).
>
> Response 2: We appreciate this suggestion. Future work will involve incorporating different metrics that capture global networks to validate GRAM's applicability. We updated the discussion part as follows:
> "Future research should explore a broader range of metrics (e.g., that capture global network properties) and evaluate the model across various graph datasets (e.g., functional vs structural connectomes) for different applications."
>
> Feedback 3: The paper compares GRAM to individual metrics but not to other existing methods for assessing graph quality or similarity, such as graph edit distance or graph kernel methods.
>
> Response 3: We acknowledge the reviewer for pointing this out. To address it, we plan to explore these comparisons in future research efforts. We mentioned that in the manuscript as follows:
> "Additionally, future work should include a comparison between GRAM and other methods for assessing graph quality."
>
> Feedback 4: The evaluation relies entirely on simulated distortions of real brain graphs.
> Response 4: We value the reviewer's feedback about that. Currently, this evaluation serves as an initial proof of concept. Future studies will expand this by incorporating GAN-generated data to strengthen the validation of our approach.
>
> Feedback 5: The paper doesn't demonstrate how improvements in graph quality assessment translate to clinical applications.
>
> Response 5: We appreciate the reviewer's remark on that. We added this sentence in the conclusion part as follows: "This approach is a significant step towards establishing a universal graph quality index for graph-based predictive studies (e.g., predicting disease progression in Alzheimer's, analyzing brain network development in infants)."
>
> Feedback 6: The paper doesn't explore how GRAM's performance changes with different combinations of metrics or types of graphs (e.g., functional vs. structural connectomes).
> Response 6: We recognize the reviewer’s point on that. This exploration is planned for future research efforts. We updated the discussion as follows:
> " Future research should explore a broader range of metrics (e.g., that capture global network properties) and evaluate the model across various graph datasets (e.g., functional vs structural connectomes) for different applications."
>
> Feedback 7: The paper doesn't discuss GRAM's computational cost compared to individual metrics.
> Response 7: We thank the reviewer's observation regarding that and we apologize for this lack of information. We have now included the following information in the GRAM section: "The training of GRAM on Google Colab took 1 hour and 14 minutes."
>
> Feedback 8: Using an MLP to optimize metric weights introduces the possibility of overfitting the specific dataset used.
>
> Response 8: We appreciate the reviewer's concern regarding that. It's important to note that we addressed this by generating 100 variations for each graph across all distortion levels, resulting in a total of 8,800 graphs. We apologize for any confusion and have clearly cited this in the manuscript:
> "We emphasize that our simulations resulted in a total of 8,800 graphs."
>
> Feedback 9: There were a few typos in the paper, e.g. "genrated" in Fig. 1 and "refrence" in Fig. 3 caption. The bottom pink banner for Fig 1 also has some distortion where text runs over weirdly.
>
> Response 9: We thank the reviewer for bringing this up, we appologize for any confusion. The mentioned typos have been corrected.
>
> Please you find here the revised version of our gram: https://drive.google.com/file/d/1vaGrsrw6g8Lscdnl5Ehbrwr6eI3y37e_/view?usp=sharing

---

### Official Review · Reviewer_tKhB · 2024-07-09

**Recommendation:** 5
**Confidence:** 4

**Clarity:**

The paper is clear and well-written, with minor areas for improvement in clarity

**Feedback:**

This paper is a good preliminary work towards universal metric. I would recommend that the paper is accepted. If accepted, it would be great if the authors rewrite the Introduction section with literature review on quantification of graph assessment. Other comments can be found in the weakness section.

**Justification:**

The paper is written clearly to understand the method. I like the idea of combining different graph metrics to create a universal one. The method flows well and easy to understand. The authors have clearly outlined the limitations and future work. They have also shared the code to reproduce the  results reported on their choice of similarity metrics.

**Reproducibility:**

Sufficient amount of details available for reproducing the main results, and open access is provided (or promised upon acceptance) to source code and/or data

**Strengths:**

The main strength of the paper is that they want to have a generalized graph assessment metric which compensates for the limitations of the existing single assessment metric. This method is simple and intuitive. The paper is written clearly. It is a decent preliminary work towards a general assessment metric.

**Summary:**

The authors propose a generalized metric to quantify the distortion between two connectivity matrix. The proposed metric is basically a weighted sum of existing similarity/distance graph metrics. They synthesized their own set of distorted graphs and trained MLP to find out the weights.

**Weaknesses:**

The main weakness of the paper is that it does not justify some of the assumptions made in the paper about for example, why is the distortion proportional to just number of edges with w != w^ and not the magnitude of the difference. The difference in the magnitude of edge weight is not amounting to the distortion. This makes me wonder of the metric is practical enough to be called a universal assessment metric. Secondly, the assumption of linearity of distortion with respect to the reported metric is not justified. Since the authors generated their own set of distortion graphs, I am curious about how this extends to GAN generated data. Therefore this paper is a preliminary work. The paper requires further refinement to establish the metric as practical. One comment of writing is that in the introduction section of the paper, the authors just reviewed different GAN methods and not different assessment metrics. If the main contribution is about a novel metric then in the introduction section, there should be a literature review about existing metrics.

---

> ### Author Response · Authors · 2024-07-14
> **Rebuttal by Authors**
>
> We thank the reviewer for the significant effort they made to provide feedback for this manuscript. We addressed each feedback as follows:
>
> Feedback 1: The paper does not justify some of the assumptions made, for example, why the distortion is proportional to the number of edges with w \neq w^ and not the magnitude of the difference. The difference in the magnitude of edge weight is not amounting to the distortion.
> Response 1: We appreciate the reviewer's remark. We added this sentence in the methods part as follows: "The objective is to detect any distortion in the generated graph, treating any alteration in edge weights as significant."
>
> Feedback 2: The assumption of linearity of distortion with respect to the reported metric is not justified.
> Response 2: We acknowledge the reviewer for pointing this out and apologize for the lack of information. To address it, we mentioned updated the GRAM section: "We opt for a linear model due to its ease of interpretation and analytical benefits [8]. In graph distortion context, the linear relationship clarifies how changes in edge weights impact overall metrics, enhancing result communication.
>
> Feedback 3: The author-generated distortion graphs need to extend to GAN-generated data.
> Response 3: We thank the reviewer for this observation. In response, this study serves as a proof of concept, and in future studies, we will incorporate GAN-generated data to further validate our approach. We added that in the limitations sections:
> "Future studies will also incorporate the use of GAN-generated data to further validate our approach"
>
> Feedback 4: The paper is preliminary work and requires further refinement to establish the metric as practical.
> Response 4: We thank the reviewer for highlighting this. We plan to incorporate more real-world data from diverse fields to validate GRAM's applicability.
>
> Feedback 5:  In the introduction section, the authors reviewed different GAN methods but not different assessment metrics. There should be a literature review about existing metrics.
> Response 5: We thank the reviewer for poiting this out. Indeed, we mentioned in the introduction some of the single metrics that are used to quantify a given aspect of the graph such as centrality and betweness. We apologize for the lack of clarity.The updated paragraph in the introduction is as follows: " However, unlike images, brain connectomes are virtually impossible to evaluate qualitatively. Instead, quantitative metrics (e.g., centrality measures [7], Aver- age Neighbor Degree [26] and Diversity Index [20]) are, thus far, a single way to evaluate the quality of the generated graphs."
>
> Please find here our revised paper: https://drive.google.com/file/d/1vaGrsrw6g8Lscdnl5Ehbrwr6eI3y37e_/view?usp=sharing

---

### Official Review · Reviewer_DGSa · 2024-07-10

**Recommendation:** 2
**Confidence:** 2

**Clarity:**

The paper is generally clear but has some clarity issues that could be addressed with moderate revision

**Feedback:**

The paper proposes a metric for evaluating the quality of generated brain connectome graphs. The proposed metric (GRAM) is composed of a model that learns an aggregation of a set of predefined known metrics. This allows the metric to be customizable, as different metrics can be employed or replaced during the learning process. The metric is tested on a set of simulated reconstructions obtained by adding various levels of distortions to ground truth graphs. Comparative experiments evaluate the correlation of the metric with the modification level.

The paper aims to provide a unified metric for evaluating the quality of generated brain graphs. This metric, GRAM, is customizable and adaptable to different graph types. Aggregating a set of metrics offers an interesting way to formulate the quality assessment of generated graphs. While this is an interesting approach, some points of the methods can be clarified.

The metric seems to be formulated considering the distortion. The proposed approach generates the distortion based on a known graph to simulate the generation process. However, some questions can be raised in regard to its application to real scenarios with a generated graph.
* Is a reference graph required? It seems that the metric compares against a reference/ground truth graph. Considering that the generated graphs in an actual implementation might not be directly associated with ground truth, what can be used as a reference graph?
* Similarly, is it necessary to know the distortion level to compute the metric? Is this parameter known in an application where the graphs were obtained with a generative model?

Similarly, some clarification can be appreciated in the Methods section. For example:
* The paper can introduce more information about the motivation description of what C represents. It is a reference metric, but how it is computed still needs clarification.
* Something similar can be commented on regarding variable B.
* The training process of the metric employs a multi-layered perceptron (MLP) and a polynomial fitting. What is the output or the main objective of the MLP, and why is it still required to use a polynomial fitting to obtain the combined metric?
* Would it be possible to provide some explanation about the surfaces Sr and Sp?

**Justification:**

The metric aim is to evaluate the quality of generated brain graphs. Since the method seems to be evaluated on simulated graphs with know distortions, its application to a real scenario might need to be discussed.

**Reproducibility:**

Sufficient amount of details available for reproducing the main results, and open access is provided (or promised upon acceptance) to source code and/or data

**Strengths:**

The paper aims to provide a unified metric for evaluating the quality of generated brain graphs.

This metric, GRAM, is not only customizable but also adaptable to different graph types.

Aggregating a set of metrics offers an interesting way to formulate the quality assessment of generated graphs.

**Summary:**

The paper proposes a metric aggregation method to define a unified metric for quality evaluation of generated brain graphs.

**Weaknesses:**

The metric seems to be formulated considering the distortion of the generated graph. The proposed approach generates the distortion based on a known graph to simulate the generation process. However, some questions can be raised regarding its application to real scenarios with a generated graph and when the distortion is potentially unknown.

Similarly, some clarification can be appreciated in the Methods section, for example, in the definitions of the variables B and C and the training process of the metric.

---

> ### Author Response · Authors · 2024-07-14
> **Rebuttal by Authors**
>
> We thank the reviewer for the significant effort they made to provide feedback for this manuscript. We addressed each feedback as follows:
>
> Feedback 1: The metric may face challenges in real scenarios where the distortion of the generated graph is unknown.
> Response 1:  We value the reviewer's feedback. To address this, users of GRAM would initially create simulated graphs with known distortion levels to fine-tune the weights of the chosen existing metrics that form GRAM. This approach ensures that the metric is approximative even when distortion levels are not explicitly known in practical applications.
>
> Feedback 2: Definitions of variables B and C need to be clarified.
> Response 2: We appreciate the reviewer's comment regarding variables B and C. According to assumption 1 in the manuscript: The vector C is a representation of linearly correlated values with distortion levels where for instance, graphs with 0.1 of distortion level their metric values are 0.9. We trained GRAM to approximate this vector C. The vector B is the output of the MLP, it contains the weight of each of the exisitng metric we chosed in this study ensuring the linearity condition stated in assumption 1. GRAM is trained to output a vector B such that AxB approximates C. We highlight the explanation in GRAM section:
> "We define C as the reference output of the metric ensuring adherence to Assumption 1. GRAM aims to find the values αj forming a vector B such that: A × B = C."
>
> Feedback 3: The training process of the metric requires more detailed explanation.
> Response 3: We acknowledge the reviewer’s comment. Accordingly, we have updated the training explanation to include more details as follows:
> "We train our model for 250 epochs, using Google Colab. For optimization, we use Adam optimize [11], with a learning rate of 0.01. We used an 80/20 split, resulting in a training set of 7,040 samples and a testing set of 1,760 samples."
>
> Feedback 4: Is a reference graph required? It seems that the metric compares against a reference/ground truth graph. Considering that the generated graphs in an actual implementation might not be directly associated with ground truth, what can be used as a reference graph?
> Response 4: We acknowledge the reviewer’s comment. The reference graphs are needed for computing our metric. Similar to any statistical measures such as MAE, MSE, etc.. GRAM requires a ground truth graph that it can use to measure the distortion level and therefore report the quality of the generated graph.
>
> Feedback 5: Ambiguity about whether the distortion level needs to be known to compute the metric.
> Response 5: We appreciate the reviewer's feedback. In response, we confirm that initially, users of GRAM would generate simulated graphs to fine-tune the weights of the selected existing metrics that constitute GRAM. Therefore, it is essential to know the distortion levels during the training process of GRAM.
>
> Feedback 6: The purpose and objective of using a multi-layered perceptron (MLP) in conjunction with polynomial fitting need to be explained.
> Response 6: We value the reviewer's input on this matter. The MLP architecture is used to find the optimal weights for the existing metrics that constitute GRAM, based on the universal approximation theory, with polynomial fitting serving as the loss function to train the MLP.
>
> Feedback 7: Explanation about the surfaces Sr and Sp is required.
> Response 7: We appreciate the reviewer's feedback regarding this. Sr denotes the reference surface, defined as the intersection of vector C with the X and Y axes. On the other hand, Sp represents the predicted surface, which is determined by the intersection of vector AxB (where B represents the output of the MLP) with the X and Y axes. Furthermore, we have revised the manuscript accordingly: "C(x) denotes the line defined by C values, while fMLP(x) corresponds to the polynomial approximation function of B (the MLP output) multiplied by A."
> "The goal is to optimize the predicted weights in B so that the vector A × B closely approximates vector C, aligning surfaces Sr and Sp."
>
>
> Please find here our revised paper: https://drive.google.com/file/d/1vaGrsrw6g8Lscdnl5Ehbrwr6eI3y37e_/view?usp=sharing

---

### Meta-Review · Area_Chair_bTFs · 2024-07-15

**Recommendation:** Accept (Oral)
**Confidence:** 5

**Metareview:**

The paper introduces GRAM, a novel and innovative method for evaluating the quality of synthetically generated brain graphs by aggregating multiple existing graph metrics. This approach is both customizable and adaptable, addressing a significant gap in the current literature by offering a unified metric for graph quality assessment. While the reviewers pointed out potential weaknesses, they acknowledged the paper's clarity, novelty, and potential for significant impact in the field. The authors have thoroughly addressed all feedback, providing detailed responses and committing to revise the work when appropriate.

---

### Decision · Program_Chairs · 2024-07-16

**Decision:**

Accept (Oral)

**Comment:**

This paper proposes a novel method (GRAM) for evaluating brain graphs and is accepted for publication considering its potential impact and the authors' commitment to address reviewer feedback in the final version.